# Is the Current Forest Management to the Northernmost Population of *Cordulegaster heros* (Anisoptera: Cordulegastridae) in Central Europe (Czech Republic) Threatening?

Otakar Holuša [1,*] , Kateřina Holušová [2] and Attila Balázs [3]

1   Department of Environmental Science and Natural Resources, Faculty of Regional Development and International Studies, Mendel University in Brno, Tř. Gen Píky 7, CZ-613 00 Brno, Czech Republic
2   Department of Forest and Wood Product Economics and Policy, Faculty of Forestry and Wood Technology, Mendel University in Brno, Zemědělská 3, CZ-613 00 Brno, Czech Republic
3   Department of Zoology, Fisheries, Hydrobiology and Apiculture, Faculty of AgriSciences, Mendel University in Brno, Zemědělská 1, CZ-613 00 Brno, Czech Republic
*   Correspondence: holusao@email.cz

**Abstract:** *Cordulegaster heros* is included in the EN category on the IUCN Red List for the territory of the Czech Republic, where it inhabits an area of approximately 100 km$^2$. All of the localities are located in the forest complex in Chřiby hills, and all of the forests fall into the category of management forests. Most of the forest stands have a high and very high degree of naturalness; they are natural forest stands. The predominant management units are Nutrient sites in middle elevations (78.2% of the area) and Oligotrophic sites in middle elevations (2.1% of the area), with stand types of *Fagus sylvatica* representing 92.5% of the area, and forest stand types of *Quercus* sp. representing 5.7% of the area. The wider alluvia in forest streams are classified as being in management unit alder and ash sites on waterlogged and floodplain soils (1.1%), with the forest stand type of *Alnus glutinosa*. The forest stands are restored by regeneration under shelterwood (97.8% of the area). The waterlogged alluvia, if a separate management unit is established for them, are restored by a regeneration by strip method. Realistically, seven factors were recorded in *C. heros* habitats, but they mostly have only point effects. Within forestry management, the factors of logging directly in the habitats and the subsequent transport of harvested timber in the habitat were recorded. The most intrusive effects were found on tractor logging roads, where fine soil washes into the stream and causes prolonged turbidity. Of the water management structures in the study area, logging roads with bridges and culverts are constructed, stream banks are reinforced with longitudinal walls at points, and stone steps in the channels are constructed only sporadically. The current forest management system can be described as a nature-friendly system, and therefore, it fully ensures the conditions for the survival of the *C. heros* population in the Czech Republic.

**Keywords:** forest management; silvicultural systems; negative factors for *Cordulegaster heros*; Cordulegastridae; Odonata; northernmost population; Czech Republic

## 1. Introduction

Forest management has historically been developed to fulfil the requirements of society with regard to timber production. Gradually, more and more requirements have been placed on forests and forest management, making forest management multifunction, as it fulfils not only production, but a whole range of non-production functions [1]. In the 20th century, the concept of "sustainable forest management" gradually emerged, which includes not only an emphasis on timber production, but also forest values [2]. In forestry practice, categories of forests other than the basic category of management forests—in the case of the Czech Republic, Forest Act No. 166/1961 [3]—have been seen with this view. Gradually, different management practices have been established for different forest

stand orientations in the form of differentiated management practices [4]. Towards the end of the 20th Century, Forest Act No. 289/1995 in the Czech Republic provided a wide range of non-productive functions with regard to the functional orientation of the forest stands [5]. Different management options for different categories and subcategories have been established so that the forest fulfils all the functions of sustainable management [6–8].

Knowing the detailed bionomics of individual species, whether they are plant or animal ones, or species of importance with regard to the biodiversity of communities or species of European importance, it is possible to determine that such forest management would ensure the maintenance of the population of these selected species, while fulfilling the other functions.

One of those species that are important for the conservation of the diversity of selected biotopes in Europe is *Cordulegaster heros*, a representative of the family Cordulegastridae of the order Odonata, which is listed as requiring protection (Annex II and IV of Council Directive 92/43/EHS). In the Red List of dragonflies of Europe, the species is classified as NT, with a stable population trend [9]. Its classification varies from country to country, especially in the northern part of its range, i.e., Austria and the Czech Republic, where it is classified as being in the category "EN: endangered" [10,11]. Although *C. heros* is one of the other species about whose exact range we do not yet have enough information, we currently know that the species exists on the Balkan Peninsula (including Greece, Albania, North Macedonia, Bulgaria, Montenegro, Bosnia and Herzegovina, Croatia, Slovenia, Serbia, north-eastern Italy, and Romania) and in Central Europe at the foothills of the Alps in eastern Austria [12–15] and also in the foothills of the Carpathians Mountains in Slovakia [16,17] and the Czech Republic [18,19]. Outside of the Carpathians Mountains, the species occurs in Ukraine [20] and in the hills in the middle of the Pannonian lowland in Hungary [21,22]. Their occurrence in the territory of the Czech Republic was discovered in 2009 [18], while the occurrence of larvae was discovered in 2011 [19]. An intensive survey in 2011–2021 found that the northernmost population of the species in the territory of the Czech Republic is isolated from the rest of the range in Austria and Slovakia [23]. This population inhabits an area of approximately 100 km² and is composed of 10 individual forest streams and brook watersheds.

Generally, the species is associated with watercourse sections in forest complexes, or at least its habitats that are bordered by accompanying forest belts [21,22,24,25]. All of its life histories take place directly in forest stands, with larvae being buried in suitable sediments of the forest watercourses, imagoes flying mostly along shaded watercourses, and females laying eggs under the screen of forest stands in the watercourse (Figure 1). Exceptionally, males may use vegetation edges or clearings or forest dumps for resting or hunting (Holuša unpubl.).

There are many factors that can threaten the population of a species [26,27], but the parameters of the forest management in forest stands can vary. The aim of this paper is to present a complex evaluation of the forest management parameters based on assessments of the forest stand condition, harvesting and regeneration systems, and interference or influence of the water flows in habitats with *C. heros*. Additionally, we assessed whether current forest management is threatening the northernmost population of *C. heros*.

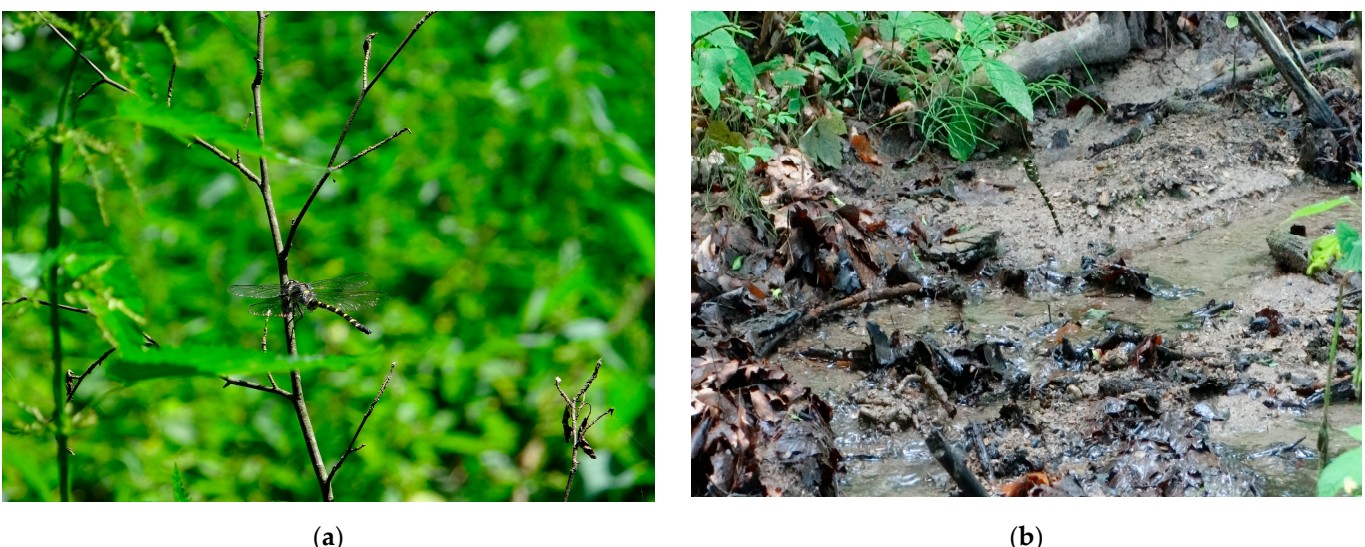

(**a**)            (**b**)

**Figure 1.** Adults of *C. heros:* (**a**) male using young tree in forests for sunbathing and resting; (**b**) female laying eggs directly in forest stands.

## 2. Materials and Methods

### 2.1. Study Area

The area covers about 100 km$^2$ in the north-eastern part of the Chřiby hills in the south-eastern part of the Czech Republic. The Chřiby hills are an important part of natural forest area No. 36: Středomoravské Karpaty (=Central Moravian Carpathians Mts.) [28,29]. The area is bounded by the link between Buchlovice village, Halenkovice village, Žlutava village, Vrbka village, Zdounky village, and Střílky village via public road No. E50 (between Střílky and Buchlovice villages) (Figure 2).

Most of the forests fall under the administration of Forest of the Czech Republic (Lesy České republiky, s.p.)  and part of the territory falls under the administration of Arcibiskupské lesy a statky, s.r.o., but only the north-eastern part of the territory and the north-eastern territory forests fall under the legal administration of private persons.

### 2.2. Evaluating the Current State of Forest Stands and Forest Management

Data from the forest management plans of the area were used to evaluate the tree species composition in terms of the spatially distributed units in the forests [30,31]. Data related to forest categories and target management units were used, including data from the Forest Management Institute [32], data of the occurrence of vegetation stages and site classification [33,34], data of the road network and road categories [35], and data of the forest stand type units [36]. Detailed management recommendations were identified for individual target site management units [29,37].

The field surveys in individual forest stands in 2017 and 2020 verified the data of the forest management plans, identified the information about the regeneration of individual stands, and determined the current state of the forest stands, especially those with *Picea abies*. A detailed survey was carried out mainly in forest stands with *C. heros*.

By evaluating the current tree species composition according to the methodology of Macků [38], information about the level of naturalness of individual forest stands was obtained. The maps were processed in ESRI 2020 ArcGIS ArcMap 10.8 software.

### 2.3. Threatening Factors in Forest Management

Factors that could affect the population of *C. heros* were divided according to their effects into factors caused by forest management, factors related to the water environment, and factors caused by construction activities. The classification of factors according to the works of Holuša et Holušová [27] and Murányi et Jović [26] was used.

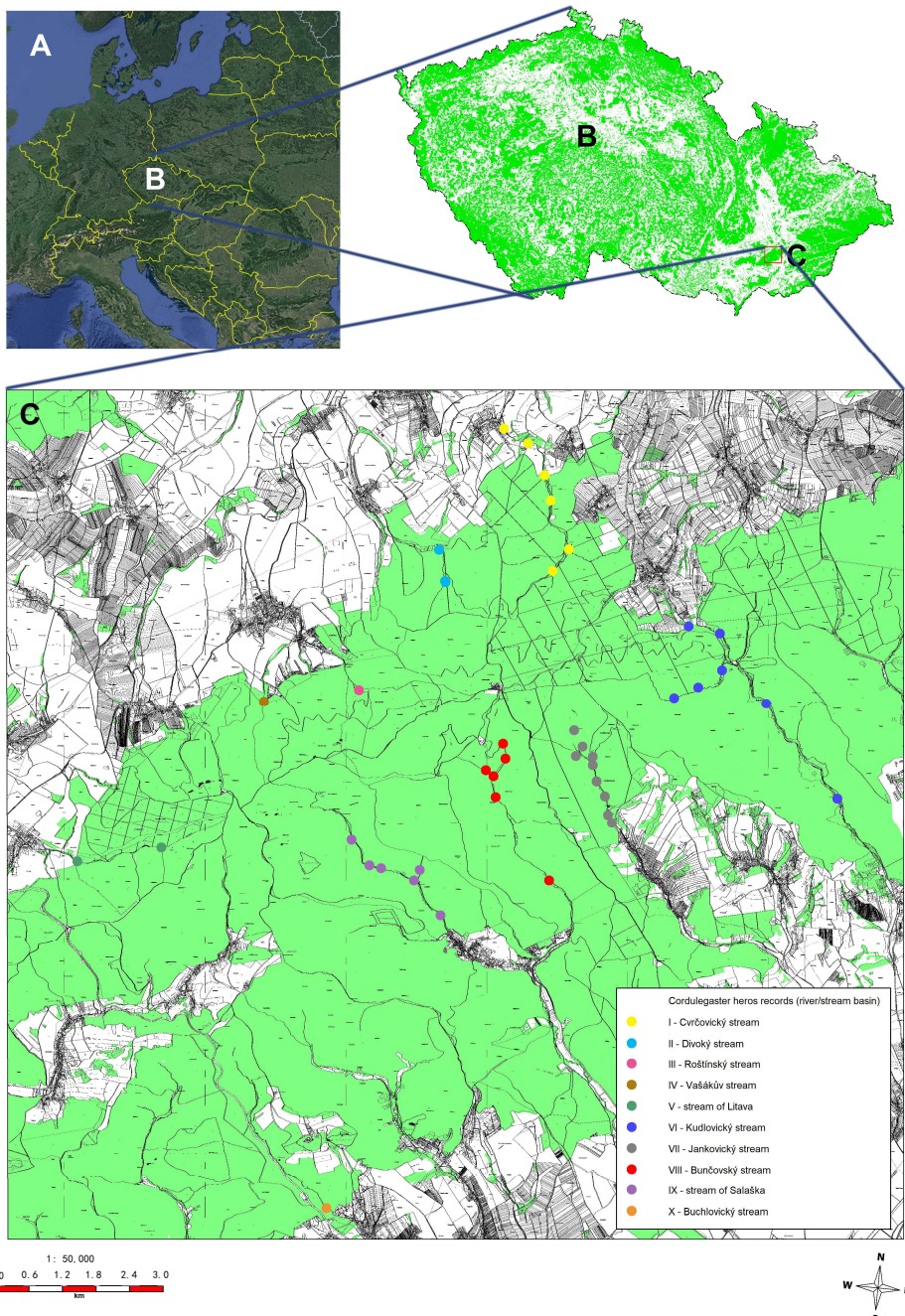

**Figure 2.** Study area of northernmost population *C. heros* in the Chřiby hills in the Czech Republic. (**A**) Map of Western, South, and Central Europe. (**B**) Map of Czech Republic with occurrence of forests (green). (**C**) Region of the Chřiby hills in the Czech Republic with *C. heros* records in individual watershed in an area of forest stands (green).

For each factor, we assessed whether it had a point effect or an area effect in a larger landscape segment and whether the factor was recorded in the *C. heros* habitat or in the forest stands in the catchment.

The assessment of *C. heros* population abundance is based on the work of Holuša et Holušová [27], which evaluated the overall abundance in individual streams. Two localities were selected to assess the influence of individual factors or groups of factors. a. Jankovický brook stream (loc. VIIc according to Holuša et Holušová [23], GPS 49°09′59.90″ N, 17°22′28.30″ E), which is influenced by the construction of a forest diversion road, and the locality is affected (from periods IV to VIII) by logging directly on the banks,

with the subsequent transport of timber towards the watercourse. b. Habešský potok stream locality (locality VIf according to Holuša et Holušová [23], GPS 49°11′07.23″ N, 17°23′40.57″ E)), which is not under any direct influence of forest management, and the forest paths that trail up to the edge of the alluvium and forest porosities have a natural character here, and the locality represents the most natural habitat. At both of the sites, transects were selected along 100 m of the stream, and the larval densities per 1 m$^2$ of suitable sediment were surveyed in month X of each year (2013, 2015, 2017, 2019, 2021, and 2022). The effect of the repair of fortification, i.e., stone riprap on the bank (a repair carried out at VI.2018; the construction lasted 1 month), was tested at the Bunčovský potok stream (GPS 49°9′20.23″ N, 17°21′20.27″ E). The densities of the larvae were measured in 2015, 2018, 2020, and 2022. At the stream of Salaška (GPS 49°9′5.03″ N, 17°19′28.32″ E), the effects of the use of fords by heavy vehicles (tractors) for logging was analysed. Logging was carried out 500 m from the watercourse, and the ford was used in period of IV. to VIII.2018. The densities of the larvae were measured each year (2017, 2018, 2019, and 2021). The densities of the larvae in the case of stone riprap repair and ford use were surveyed on a 100 m transect above the activity site and on a 100 m transect directly below the activity site. We tested the effect of the selected anthropogenic interventions on the density of the larvae in suitable sediments in the studied streams. All of the localities were analysed using Pearson's X$^2$ test in the R programme [39].

## 3. Results

### 3.1. State of Forest Stands

The study area lies in the upland region, which in terms of natural vegetation is classified as being in the oak–beech and beech vegetation tiers [40], i.e., that the dominant communities are *Querceto-fageta* s.lat and *Fageta abietis* s.lat. [41]. The reconstruction of the natural tree composition according to the forest site type complexes [8,41] would be dominated by European beech (*Fagus sylvatica*) (45.8%), Pedunculate oak (*Quercus robur*) and Sessile oak (*Q. petraea*) (34.8%), Small-leaved lime (*Tilia cordata*) and Large-leaved lime (*T. platyphyllos*) (10.2%), Maples (*Acer* sp.) (without species differentiation) (2.8%) and Silver fir (*Abies alba*) (1.5%), with other species presenting less than 1% of it (Table 1). By comparing the current composition and the natural composition, it is clear that the compositions are different, with *Fagus sylvatica* (now 28.8%), *Quercus robur* and *Q. petraea* being better represented (now 24.0%) than European hornbeam (*Carpinus betulus*) (now 11.3%) and Scotch pine (*Pinus sylvestris*) (now 5.7%) are. A comparison of the current and potential compositions of the individual units in terms of their spatial distribution in the forest according to the naturalness assessment [38] shows that most of the stands have a high (grade 4) and very high (grade 5) degree of naturalness, i.e., they are natural stands. From the group of geographically non-native tree species (i.e., non-native to the natural forest area of the Středomoravské Karpaty (=Central Moravian Carpathians)), there are Norway spruce (*Picea abies*) (9.9%), European larch (*Larix decidua*) (7.0%) (Table 1), and only a single admixture of Black locust (*Robinia pseudoacacia*) on the southern slopes of the area.

Overall, the forest stands show an average stock of 253 m$^3$.ha$^{-1}$, which is approximately the same as the average stock, 267 m$^3$.ha$^{-1}$, for the whole of the Czech Republic [42]. The average annual harvest is 5.05 m$^3$.ha$^{-1}$ compared to the average for the country, which is 11.29 m$^3$.ha$^{-1}$, and the average forest stand in the area has a life span of 122 years, with the average life span for the country being 115 years [42] (Table 2). The forest stands are composed of *Fagus sylvatica* (100%) or *F. sylvatica* with admixtures of *Tilia cordata* and *Ulmus glabra*; the areas represent very productive stands, and since they have an age of 140–187 years, they are able to have a stock of 800–1000 m$^3$ [43].

**Table 1.** Tree composition of forests in the Chřiby hills within the natural forest region of the Středomoravské Karpaty (Central Moravian Carpathians).

| Tree Species [1] | Present Representation (%) | Natural Representation (%) | Optimized Recommended Representation (%) |
|---|---|---|---|
| European beech (*Fagus sylvatica*) | 28.8 | 45.8 | 44.1 |
| Pedunculate oak (*Quercus robur*) Sessile oak (*Quercus petraea*) | 24.0 | 34.8 | 25.8 |
| Small-leaved lime (*Tilia cordata*) Large-leaved lime (*Tilia platyphyllos*) | 4.1 | 10.2 | 1.5 |
| European hornbean (*Carpinus betulus*) | 11.3 | 4.8 | 3.5 |
| Sycamore (*Acer pseudoplatanus*) Norway maple (*Acer platanoides*) Field maple (*Acer campestre*) | 1.0 | 2.8 | 5.3 |
| Silver fir (*Abies alba*) | 0.1 | 1.5 | 0.6 |
| Common ash (*Fraxinus excelsior*) | 1.1 | 0.6 | 1.1 |
| Black alder (*Alnus glutinosa*) | 0.6 | 0.6 | 0.5 |
| Scotch pine (*Pinus sylvestris*) | 5.7 | 0.4 | 0.9 |
| Silver birch (*Betula pendula*) | 4.6 | + | 0.7 |
| Norway spruce (*Picea abies*) | 9.9 | 0 | 3.1 |
| European larch (*Larix decidua*) | 7.0 | 0 | 8.3 |

[1] species ranked according to their distribution in the natural composition; +—occurrence only in single specimens.

**Table 2.** Basic dendrometric indicators of forest stands in the study area.

| Indicator | Value |
|---|---|
| Average stock of forest stands | 253 $m^3.ha^{-1}$ |
| Average annual production | 5.05 $m^3.ha^{-1}$ |
| Average rotation period | 122 years |

The age distribution of the forest stands is unbalanced, with forest stands that are aged between 51–60 years and 101–110 years being significantly higher represented than the normally distributed age classes are (Figure 3), which is due to the historical development of afforestation and forest use in the area. In contrast, forest stands that are aged from 0 to 50 are less well represented, having significantly less representation than would be the case in the "normally distributed age classes".

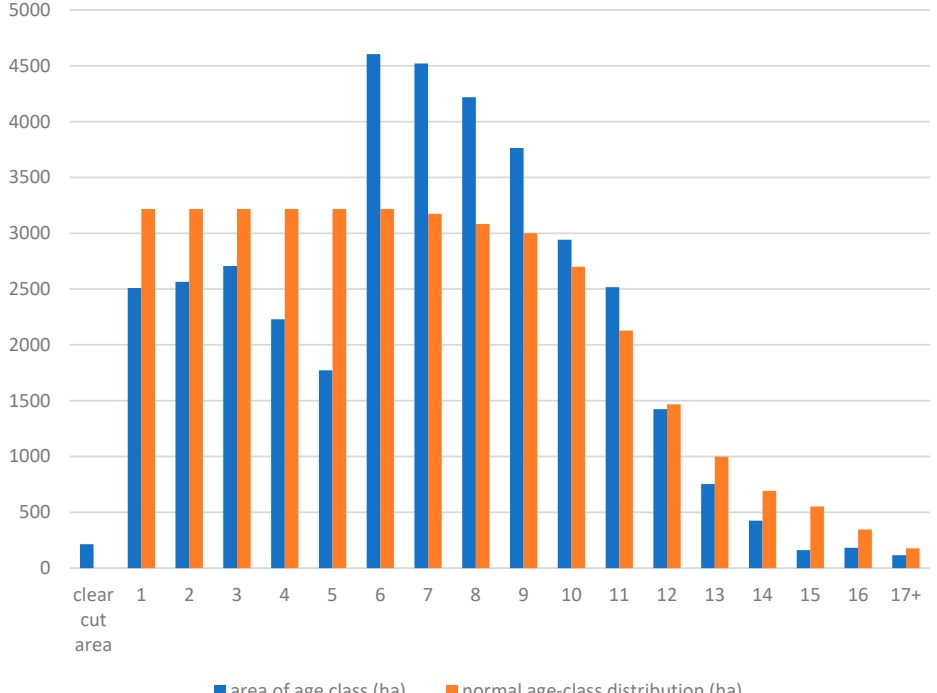

**Figure 3.** Age structure of forest stands (age classes) in the study area with a comparison of the normally distributed age classes.

### 3.2. Forest Management System

#### 3.2.1. Categories of Forests and Forest Management Units

The whole area falls into the category of management forests, and only the rock outcrops on the main ridges could be classified as protective forests, but due only to their occurrence, no units that are spatially distributed in the forest are placed in this category [32]. In terms of management, most of the areas, including habitats with *C. heros*, are classified as being site management units (a site management unit (SMU) is defined based on the site conditions [29], the altitudinal position and the trophic and hydric conditions, and these are represented by two-digit numeric codes: SMU 45 is nutrient sites of middle elevations (78.2%), SMU 41 is exposed sites of middle elevations (12.2%), and SMU 25 is nutrient sites of lower elevations (4.7%)). The actual stream alluvium, if it is spread wide and is spatially distributed, is classified as SMU 19, natural floodplain sites (0.7%), and SMU 29, alder and ash sites on waterlogged and floodplain soils (1.1%) (Figure 4a, Table 3). The site management unit is supplemented by the current type of forest stand, thus giving a specific management unit for which the basic parameters of silvicultural system are determined (the two-digit code is supplemented by the code of the predominant tree species-site management unit with forest stand type has a three-digit code f.e. 456). In the study area, the forest stand type *Fagus sylvatica* refers to forest stands dominated by *Fagus sylvatica* or those in which monocultures of *Fagus sylvatica* are predominant, i.e., 92.5% of the area in total (Figures 4b and 5a). *Quercus* sp. forest stand types are account for 5.7% of the area, and only 1.1% of it is classified as the forest stand type *Alnus glutinosa* (Figure 4b, Table 3).

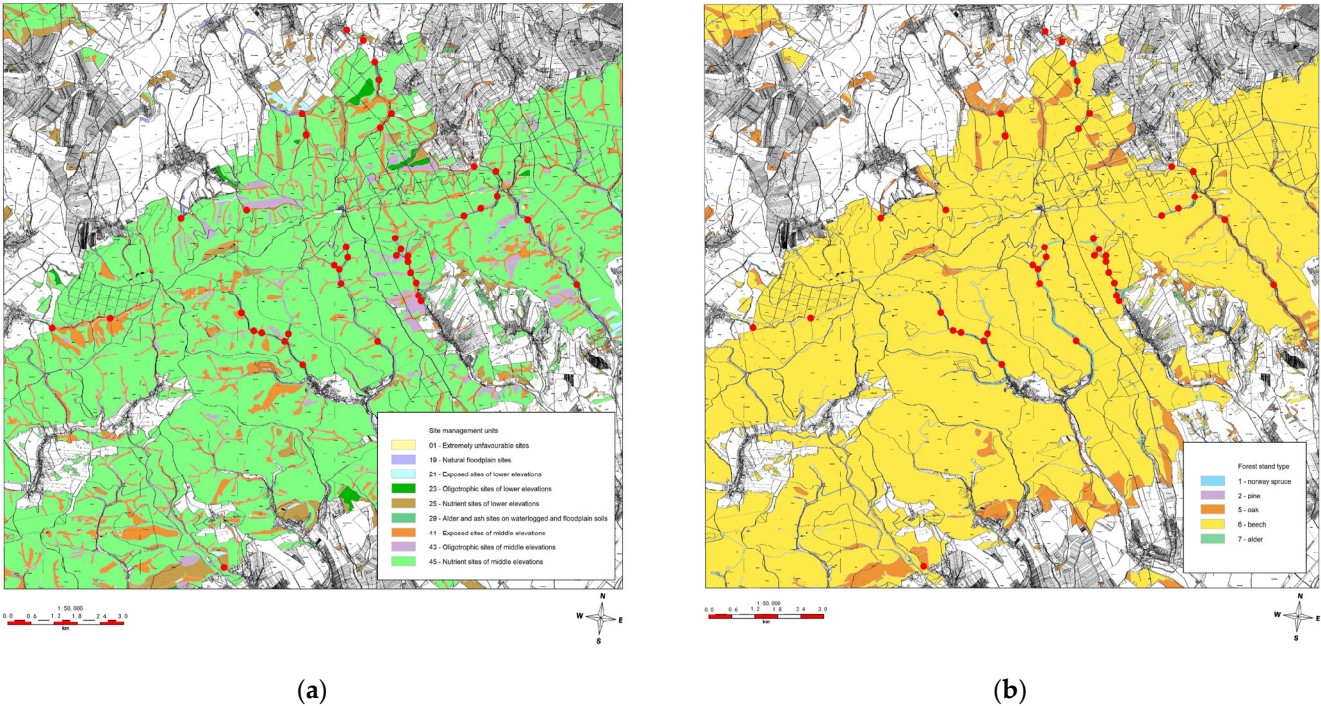

(**a**)                                        (**b**)

**Figure 4.** Area of *C. heros* occurrence with units of forest management: (**a**) management site units; (**b**) forest stand types.

The detailed silvicultural system with a temporal adjustment is informed by the site management units with different forest stand types (SMUS), which thus represents the basic unit of management, and each spatially distributed unit in the forest has its own classification [6,29,37]. The most important basic parameter is the silvicultural system. According to Forest Act No. 289/1995 [5], four ways of achieving the goals of the silvicultural system of the regeneration of forest stands are applied in the Czech Republic [29,44,45]:

a. regeneration under shelterwood, in which forest regeneration takes place under the

protection of the harvested stand, b. regeneration by strip, in which forest regeneration takes place on a continuously harvested area, whose width does not exceed the average height of the harvested stand, c. the clear-cut method of regeneration, in which reforestation is carried out on a continuously harvested area that is wider than the average height of the harvested stand, however, the size may not exceed 1 ha, and d. the selection method of regeneration, in which harvesting for reforestation and silviculture is not differentiated in time and space and is carried out by selecting individual trees or groups of trees in the stand area.

**Table 3.** Overview of management files for forest stands in the catchment area of watercourses with *C. heros* in Chřiby hills.

| Site Management Units (Code/Name) | | Site Management Units with Forest Stand Type (Code/Name) | | Representation (%) | Silvicultural System * | Rotation Period (Year) | Regeneration Period (Year) |
|---|---|---|---|---|---|---|---|
| 19 | Natural floodplain sites | 195 | oak | 0.7 | N, P | 160 | 30 |
| 21 | Exposed sites of lower elevations | 215 | oak | 0.4 | pN, P | 130 | 30 |
| 23 | Oligotrophic sites of lower elevations | 235 | oak | 0.6 | P, p(n)N | 130 | 30 |
| 25 | Nutrient sites of lower elevations | 255 | oak | 4.7 | nP, pN | 140 | 30 |
| 29 | Alder and ash sites on waterlogged and floodplain soils | 297 | alder | 1.1 | pN | 80 | 20 |
| 41 | Exposed sites of middle elevations | 416 | beech | 12.2 | P, pN | 130 | 30 |
| 43 | Oligotrophic sites of middle elevations | 436 | beech | 2.1 | P, pN | 130 | 30 |
| 45 | Nutrient sites of middle elevations | 456 | beech | 78.2 | P, pN | 120 | 40 |
| 01 | Extremely unfavourable sites | 016 | beech | 0.01 | V | ∞ | ∞ |

* N—regeneration by strip; P—regeneration under shelterwood; V—selection system of regeneration; p—initial use of regeneration under shelterwood in front of the main restoration area; n—initial use of regeneration by strip in front of the main restoration area.

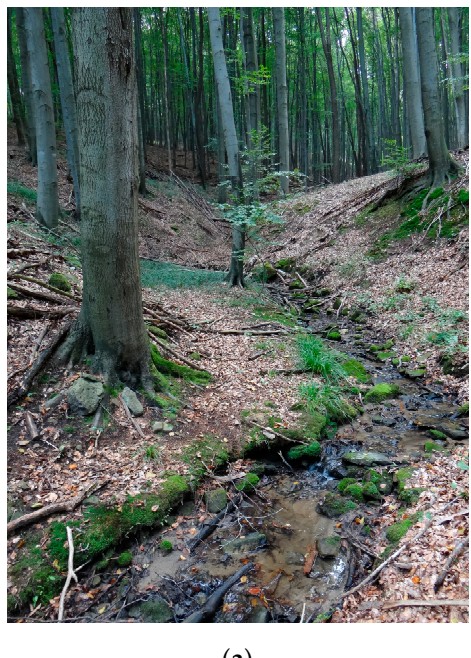

(**a**)

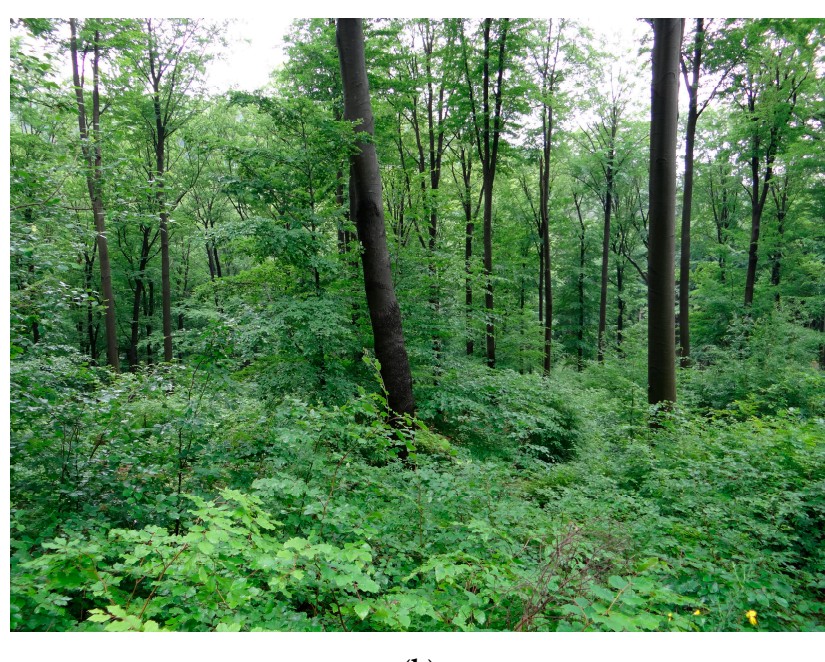

(**b**)

**Figure 5.** Forest stand type with dominancy of *Fagus sylvatica*, forest management unit 456: (**a**) detail of stand structure; (**b**) regeneration of stands dominated by *F. sylvatica* using shelterwood.

SMUS have basic management parameters: a. the management restoration method, b. the rotation period of the forest stands, and c. the regeneration period of the stands [29]. The area is clearly dominated by SMUS when the silvicultural method of regeneration under shelterwood is used, i.e., SMUS 456 (Figure 5a,b), 436, 416, and 215 (Table 3), i.e., 97.8%

of the area. Regeneration under shelterwood is also used in SMUS 195 and 215 (Table 3), i.e., 1.1% of the area. A wide spread of alluviums in a *C. heros* habitat, if separate SMUS had been established for them, are classified as SMUS 195 or 297 (Figure 6a,b), where the silvicultural method of regeneration by strips is applied (Table 3), with the possible choice to initially use the regeneration under shelterwood technique for the front of the main restoration area to restore the stands with regard to their species diversity. Exceptionally, in *Fagus sylvatica*-dominated stands, the clear-cut method is also used in the case of over-aged stands that are decaying, e.g., due to bark scar. However, these are rather portions of stands or small stands where natural regeneration cannot be expected due to rich plant vegetation stands.

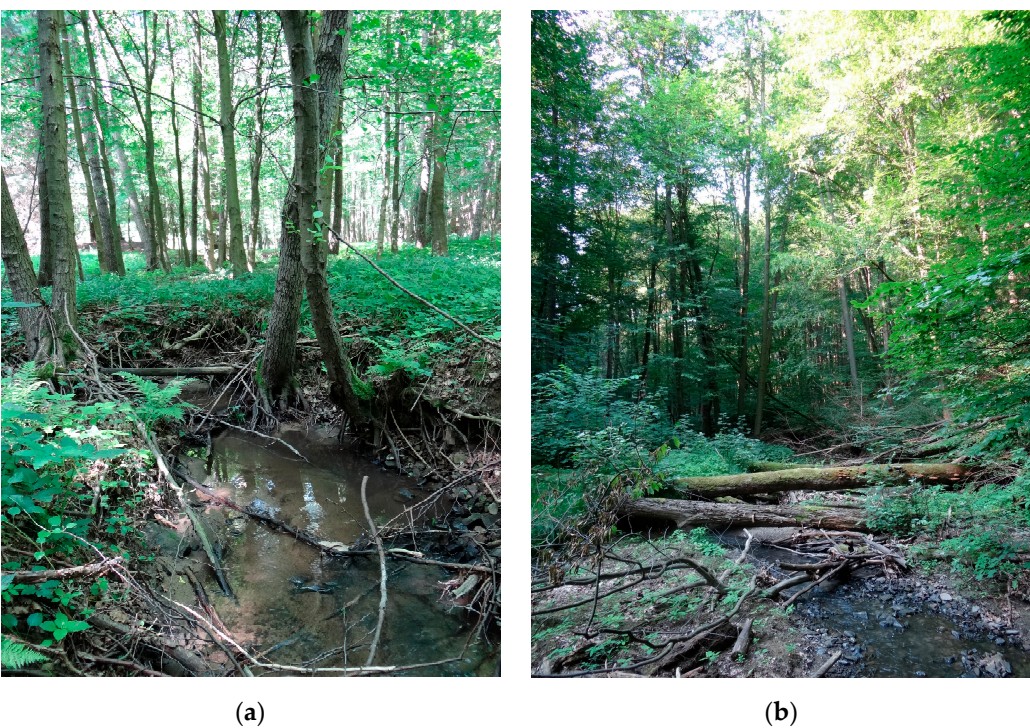

| (**a**) | (**b**) |

**Figure 6.** Floodplain forests on wider alluvium with *C. heros,* forest management unit 297: (**a**) forest stand with *Fraxinus excelsior* and *Alnus glutinosa;* (**b**) forest stand with vertical structure of *Acer pseudoplatanus*, *F. excelsior*, *A. glutinosa*, *Fagus sylvatica,* and *Carpinus betulus*.

For the SMUS, the rotation period parameter is determined with respect to the tree composition (or dominant tree species), which in the case of the study area, reaches 120–140 years for *Fagus sylvatica* stands, 160 years in the case of the floodplain sites dominated by *Quercus robur*, and 80 years in SMUS dominated by *Alnus glutinosa*. This means that mature forest stands are always restored from the stem to over-aged stem stages. The regeneration period in the case of the SMUS (456, 436, 416, 195, 215, 235, and 255) with *Fagus sylvatica* or *Quercus* sp. is in the range of 30–40 years, and only in the case of SMUS 297 dominated by *Alnus glutinosa* is 20 years chosen with regard to the ecological strategy of *Alnus* sp.

There are currently residues of forest stands dominated by *Picea abies* (Figure 7a) (which would normally be classified as SMUS 451, 431) that are subjected to "spruce dieback" and are decaying, so these stands are being restored by the clear-cut silvicultural method (Figure 7b). The following stand with a more natural tree composition was restored: a mixture of deciduous trees dominated by *Fagus sylvatica* and *Quercus* sp.

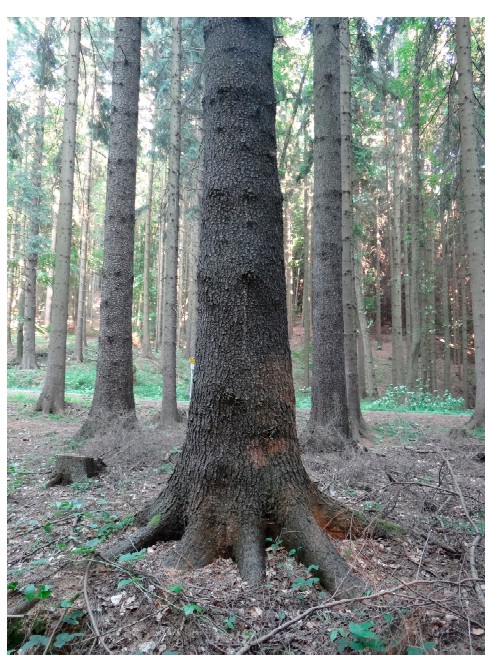 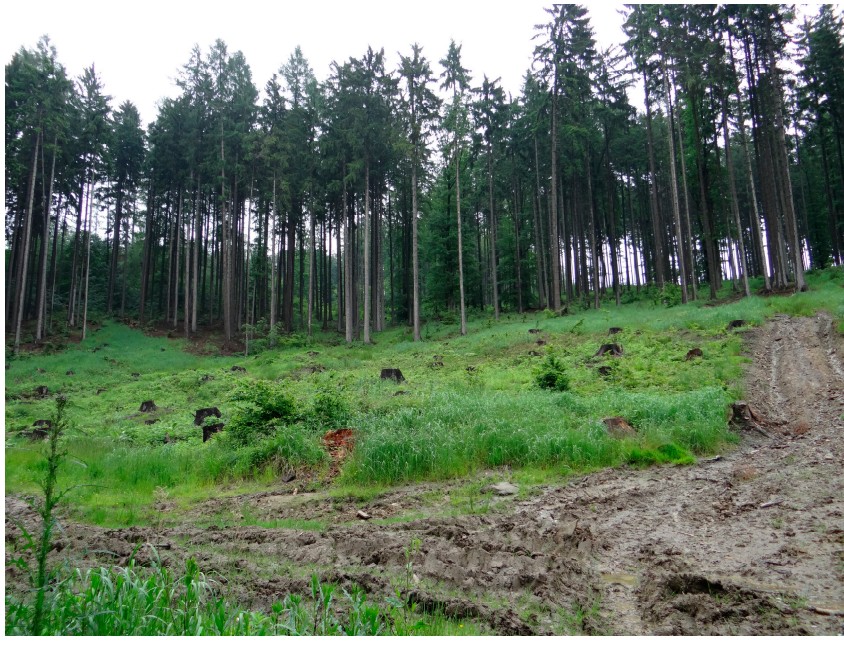

(**a**)  (**b**)

**Figure 7.** Forest stand with dominancy of *Picea abies* with occurrence of *C. heros,* forest management unit 451: (**a**) detail forest stand structure in alluvium of stream; (**b**) clear-cut regeneration of forest stand with dominancy of *P. abies*.

The resulting clearings in the actual habitat of *C. heros* may also be "unshaded" (Figure 8a), but due to the limitation of the size of the clearing area, this will only occur for a maximum of 50 m of stream length. When one is restoring the forest cover in the actual alluvium, the trees are felled directly next to the stream, thus temporarily "uncovering" and illuminating the stream itself (Figure 8b). In the watershed of the Kudlovický potok stream, as a result of the decay of the *Picea abies* groups (Figure 9), 5.73 ha of clearings were created in 2020, 4.44 ha of them were created in 2021, and 6.87 ha of them were created in 2022. Hollows are mostly caused by the accidental harvesting of dead *Picea abies* groups; no hollows were created in the habitats of *C. heros* in 2020–2022 in the Kudlovický potok catchment.

### 3.2.2. Forest Engineering Constructions

Due to the category of management forests in the area, the whole area is accessible by vehicles for the transport of harvested timber. In the study area, three categories of roads have been built [46,47]: a. main forest roads in the first category, which are forest logging roads, usually single-lane ones, allowing year-round operations due to their spatial arrangement and ability to support the transport of technical equipment; b. main forest roads in the second category are single-lane forest logging roads, allowing at least seasonal operations due to their spatial arrangement and ability to support the transport of necessary technical equipment, on which winter maintenance is not carried out; c. tractor logging roads, which are used for the skidding of timber, are passable by forestry tractors and special exporting and approaching vehicles, and where the limiting factor is the bearing capacity of the bedrock and its susceptibility to erosion as it has no road surface.

The entire area of the Chřiby hills is accessible by forest roads (Figure 9), with a modelled density of 22.5 m.ha$^{-1}$ [47]. The road network has mainly been constructed along ridges and in the valley positions, but also in stages. The most prominent ones are the first or second category main logging roads in stream valleys, which had been built parallel to streams and often affect the stream bank, and often the stream bank is unilaterally screened (Figure 10a). These roads have been constructed in all of the valleys of the area of interest.

Where a road alternates between the right and left banks of a watercourse, bridges or pipe culverts have been constructed. These haul roads are connected to tractor roads used for transporting timber from the harvested stand (Figure 10b). Often, the tractor roads cross the watercourse by a ford, so the timber is dragged across the watercourse itself. Fords occur at a rate of about 1 ford per 2 km of stream length.

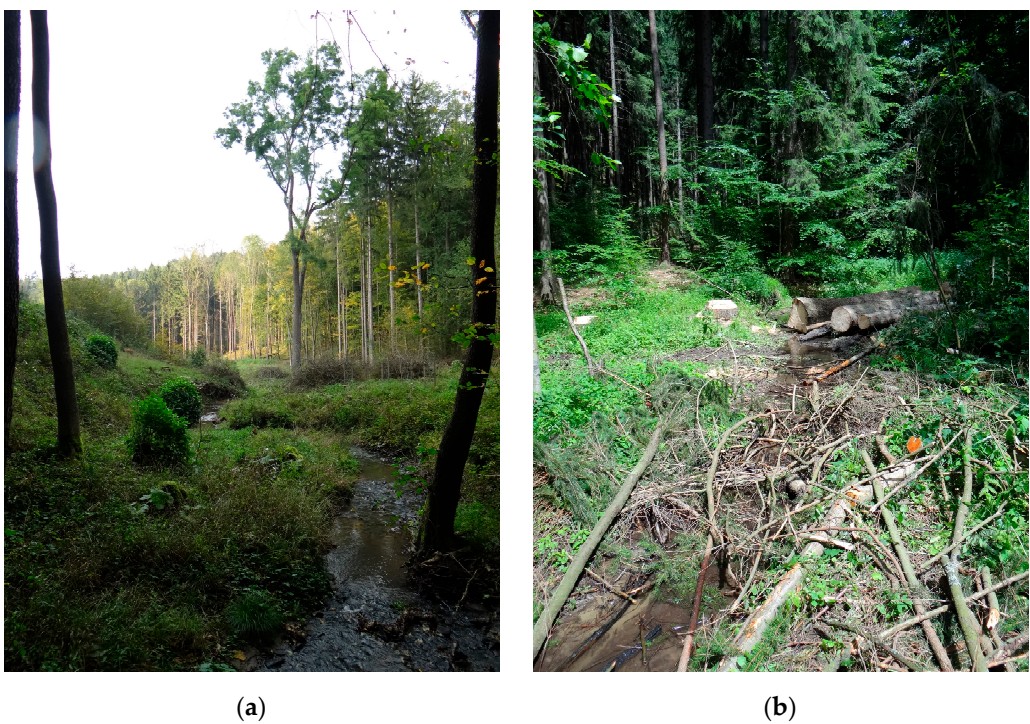

(**a**)                                                     (**b**)

**Figure 8.** Harvesting of forest stands in habitats of *C. heros*: (**a**) clear cut on the banks of stream; (**b**) individual harvesting in the habitat and also in the stream.

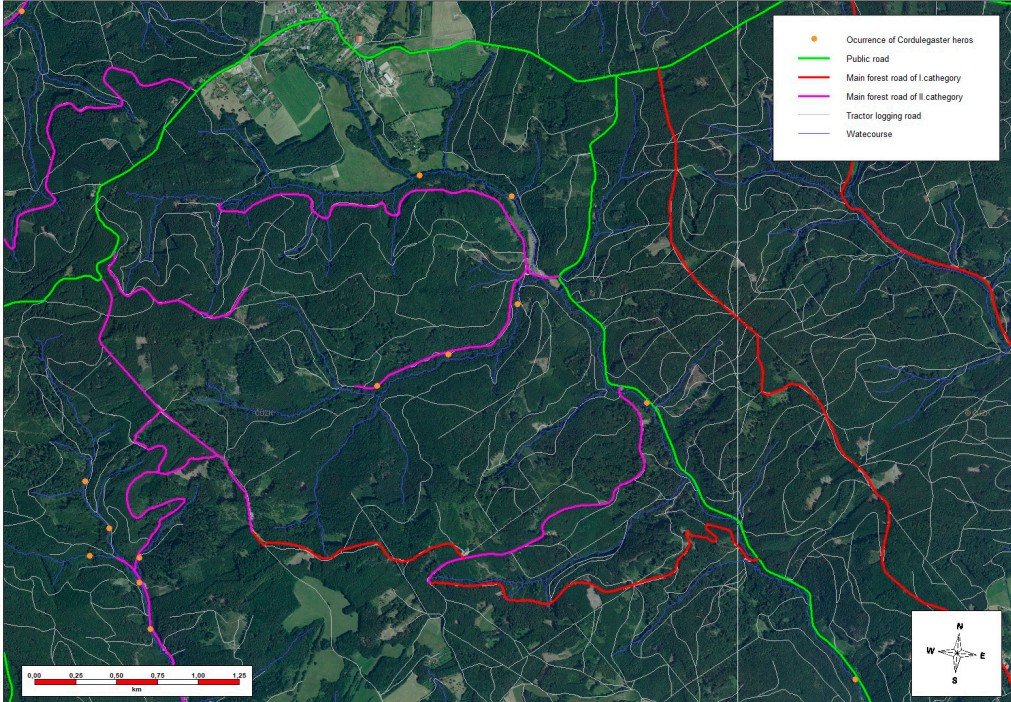

**Figure 9.** The northern half of the Kudlovický stream watershed in Chřiby hills (Czech Republic) with the network of forest transport roads and illustration of clearing and regeneration of forest stands.

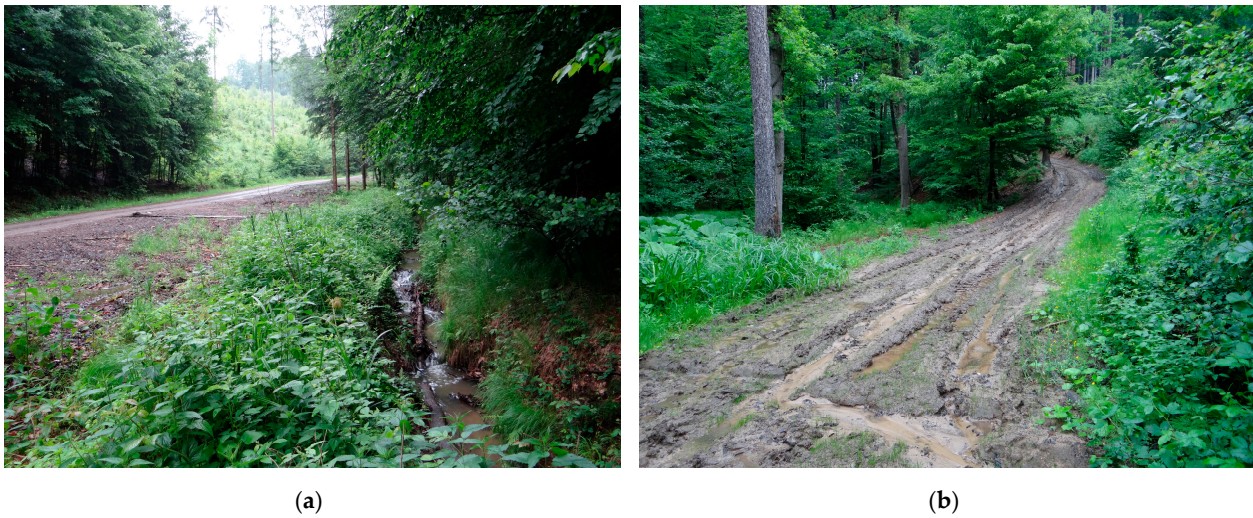

(**a**)　　　　　　　　　　　　　　　　　(**b**)

**Figure 10.** Forest engineering structures serving forest management in habitats of *C. heros*: (**a**) main forest road in the first category; (**b**) tractor logging road.

In the watercourses themselves, there are occasional forestry structures in the form of a. longitudinal fortification of the banks or b. stone steps. Stream bank reinforcements in the form of stone riprap (Figure 11a) occur where the watercourse disturbs the body of the main logging road, averaging about 30 m of the reinforcement per 1 km of stream length. In some places, after the construction of the main logging road, the watercourse bed has been straightened, and it was subsequently necessary to level the water stream gradient by constructing a wire-stone sill (Figure 11b). Stone sills occur at a rate of approximately one sill per 6 km of stream. These steps experience significant fine sediment deposition, occasionally to support *C. heros* larvae occurrence.

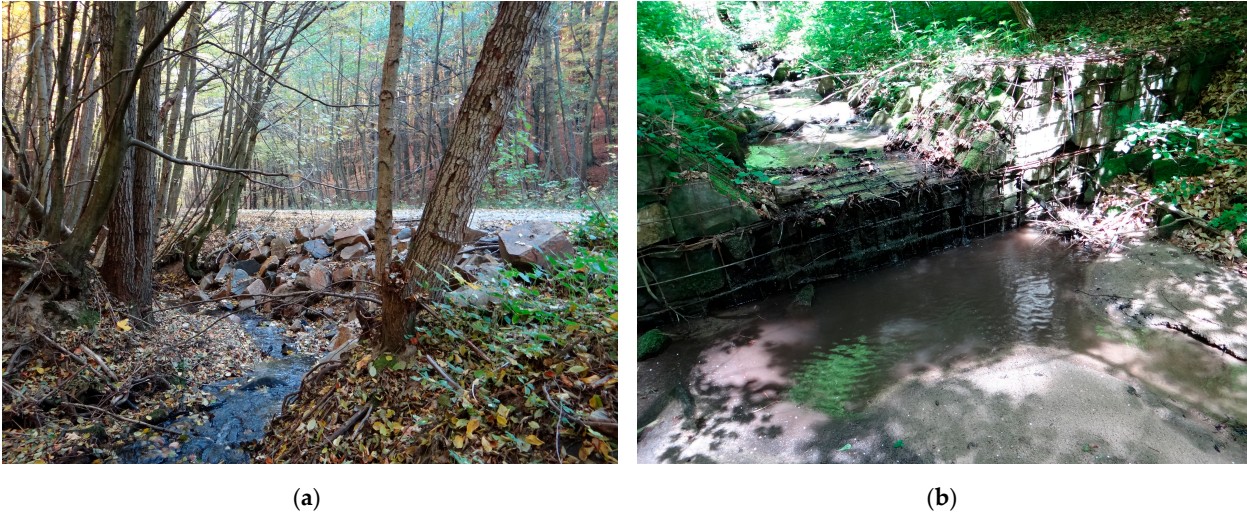

(**a**)　　　　　　　　　　　　　　　　　(**b**)

**Figure 11.** Forest engineering structures in watercourses with the occurrence of *C. heros*: (**a**) reinforcement of the banks with stone riprap; (**b**) wire-stone sill.

### 3.3. Threatening Factors on the Population of C. heros

Table 4 lists all of the real and potential factors that may affect the *C. heros* population. Realistically, seven factors have been recorded in *C. heros* habitats, but they mostly have only point effects. Within the group of forest management factors, deforestation factors were recorded, but in the study area, only those in the form of temporary deforestation during the reforestation of forest stands were recorded. These are not permanent deforestation factors. In addition, the harvesting of forest stands directly in the habitat and the

subsequent transport of the harvested timber from the habitat were also recorded, with short-term damage to the streambed caused by forest tractor movements and the transport of timber through the streambed (Figure 8b), additionally, point effects on the streambed also occurred when the harvested timber was transported over fords. The most intrusive effects were found for the tractor logging roads (Figure 10b), which are used during periods of higher rainfall (rain), resulting in fine soil being washing into the stream and longer term turbidity. Of the water management structures in the study area, forest roads with bridges and pipe culverts have been constructed, stream banks have been reinforced with longitudinal embankments at points, and stone steps in the streambeds have been constructed only sporadically. All of the structures were built 30–70 years ago, no new road or step construction is being planned, and there are ongoing spot repairs to the embankments on public roads. There is a plan to build one small reservoir in the stream of Salaška valley [48], but the building plan was rejected.

**Table 4.** Overview of potential and threatening factors or effects influencing the population of *C. heros* in Chřiby hills.

| Threatening Factor | | Effect | | Findings | |
|---|---|---|---|---|---|
| | | **Point Wise** | **Area Wise** | **in Habitat of *C.heros*** | **in the Watershed** |
| **Forest management activities** | | | | | |
| a. | Harvesting and removal of all trees in the alluvium or in the streambed (long-term deforestation) | + * | - | + * | + * |
| b. | Extensive change in tree species composition of forest stands; significant change in the degree of naturalness of forest stands | - | - | - | - |
| c. | Extensive introduction of introduced (geographically non-native) tree species | - | - | - | - |
| d. | Thinning of the forest belt along the watercourse outside the forest complex without restoration | - | - | - | - |
| e. | Transport of timber over the stream bed | + | - | + | - |
| f. | Timber transport across stream beds and fords by forestry tractors | + | - | + | - |
| g. | Large soil preparation and other types of soil erosion in forestry | - | - | - | - |
| h. | Use of natural fertilizers in forestry | - | - | - | - |
| i. | Use of synthetic fertilizers in forestry, including liming | - | - | - | - |
| j. | Use of chemical plant protection products in forestry | - | - | - | - |
| **Change of water character** | | | | | |
| a. | Chemical or biological pollution of groundwater and surface water | - | - | - | - |
| b. | Modification of hydrological conditions or physical change in water surfaces and drainage for forestry improvement purposes | - | - | - | - |
| c. | Flushing of fine ground by concentrated runoff from surrounding (harvested and restored) forest stands | + | - | + | - |
| **Forest engineering structures** | | | | | |
| a. | Transverse objects; sills | + | - | + | - |
| b. | Longitudinal reinforcement of the streambed banks | + | - | + | - |
| c. | Water reservoirs | - | - | - | - |
| d. | Forest logging roads in close to the streambed | + | + | + | + |

Note: "+" means factor detected, "-" means factor not detected; *—harvesting only for the purpose of reforestation of forest stands.

### 3.4. Influence of Factors to Population of Cordulegaster heros

The larval densities of *C. heros* per 1m² of suitable sediment in individual years at the Jankovický potok stream site (stream affected by active forestry activity) and the Kudlovický potok stream site (natural stream without forestry activity) are shown in Figure 12. The larval densities per 1 m² of suitable sediment in individual years at the site that experienced the effect of the repair of the reinforcement on the bank of the stream are shown in Figure 13a, and changes in the same parameter with the effect of the use of fords to move across the stream are shown in Figure 13b. The effect of the tested anthropogenic activities (i.e., forest road construction and the transport of logs on the stream compared to that on a

natural part of stream, the influence of the repair of watercourse fortification by stone riprap, and the use of fords by heavy vehicles to transport logs across the stream) have not had a significant effect on the larval densities of *C. heros* in the studied habitats (effect of using forest road and timber cutting with logging: X-squared = 0.74757, df = 5, *p*-value = 0.9803; repair of watercourse fortification: X-squared = 0.54074, df = 3, *p*-value = 0.9099; use of ford: X-squared = 0.55188, df = 3, *p*-value = 0.9074).

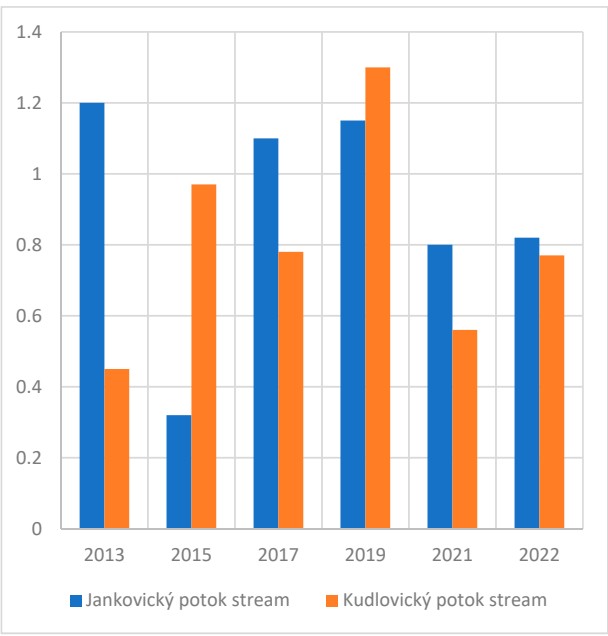

**Figure 12.** Comparison of larval densities of *C. heros* per 1 m$^2$ of suitable sediment (y-axis) in individual years at the Jankovický potok stream site (stream affected by active forestry activity) and the Kudlovický potok stream site (natural stream without forestry activity).

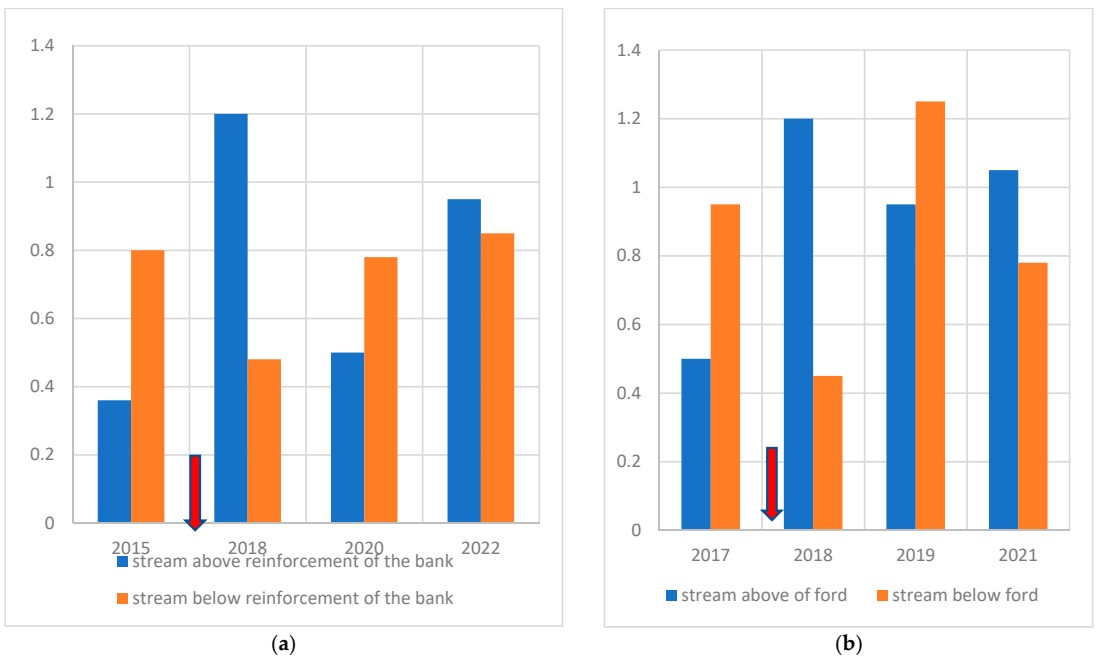

**Figure 13.** Comparison of larval density *C. heros* per 1 m$^2$ of suitable sediment (y-axis) in different years at sites: (**a**) with the effect of repair of reinforcement (site Bunčovský potok stream); (**b**) with the effect of using of ford across stream (site stream of Salaška); red arrow shows the occurrence of the phenomenon.

## 4. Discussion

The species *C. heros* is included in European legislation, it is included in *Annex II: Species of fauna and flora of Community interest whose conservation requires the designation of special areas of conservation* and *Annex IV: Species of fauna and flora of Community interest requiring strict protection* in Council Directive 92/43/EEC from 21 May 1992 on the conservation of natural habitats and of wild fauna and flora. According to the IUCN Red List of Threatened Species, it is classified as NT: Near Threatened, with a declining population trend [9]. In the Red List of the dragonflies in Europe, the species is classified as NT, with a stable population trend [49]. Its classification varies from country to country with respect to its range and population density, and of course, the state of the knowledge (Table 5). In the northern part of its range, i.e., Austria and the Czech Republic, it is placed in the category "EN: endangered" [10,11]. The neighbouring countries, i.e., Slovakia and Ukraine, have not yet classified the species with regard to its "discovery" or absence of Red List categorization [50–53]. In the centre of the range, i.e., Slovenia and Serbia, the species is included in the category "VU: vulnerable" [54,55], whereas in Croatia or Greece, the species is not included [56,57]. This clearly shows that the species is included in the national Red Lists in most of the countries where it occurs, and this determines its importance within the dragonfly fauna and its conservation.

**Table 5.** *C. heros* and its national conservation status in national Red Lists in its distribution range.

| Country | Category of Red List | Source |
|---|---|---|
| Czech Republic | EN | [11] |
| Austria | EN | [10] |
| Slovakia | Uncategorized | [50] |
|  | VU | [51] |
| Ukraine | Uncategorized | [52,53] |
| Italy | NT | [58] |
| Slovenia | VU | [54] |
| Croatia | Uncategorized | [56] |
| Serbia | VU | [55] |
| Bosnia and Herzegowina | NT | [59] |
| Romania | VU | [51] |
| Bulgaria | EN | [60] |
| Montenegro | ? [1] | - |
| Albania | ? [1] | - |
| Northern Macedonia | ? [1] | - |
| Greece | Uncategorized | [57] |
| Europe | NT | [9] |
| East Mediterranean region | NT | [61] |

[1] it is not known whether the Red List of Odonata has been elaborated.

The population of *C. heros* in the Chřiby hills area in the Czech Republic inhabits a forest complex with a species composition that is dominated by natural tree species, which are *Fagus sylvatica*, *Quercus petraea*, and *Carpinus betulus*, etc., the stands, therefore, have a high ecological stability, and there is a very low probability that they would undergo extensive decay which would cause the extensive deforestation of the entire area [62]. *Picea abies* stands are undergoing a major change, and their populations are declining in the Central European region [63–65], and in some areas, especially where *Picea abies* is geographically non-native, their populations will reduce in size or disappear. The collapse of these stands in the area of the Chřiby region creates clear cuts, i.e., temporary deforestation. According to the current Forest Act No. 289/1995 [5], these areas must be reforested within 2 years. Locally, the dieback of *Fraxinus excelsior* by *Hymenoscyphus fraxineus* has also been recorded in the study area, however, there was no extensive decay of the tree canopy as *Fraxinus excelsior* occurs only as an admixture. Newly introduced geographically non-native tree species (e.g., *Pseudotsuga menziesii*) are not introduced by current management projects. The only danger is the spontaneous spread of *Robinia pseudoacacia* (Holuša nepubl.), and in

the case of the Chřiby hills, the species has only been found in the foothills in the stand margins, and it is not likely to spread, perhaps even along the roads, into the centre of the Chřiby, and therefore, significantly affect the habitat condition of *C. heros*. The goal of management in the future is to try to change the tree species' representation (Table 1: optimised recommended representation), with the main tree species *Fagus sylvatica* and *Quercus robur* and *Q. petraea*, etc., approaching the ideal proportional representation in the natural composition. The only major difference from the natural composition is the retention of a relatively high proportion of *Larix decidua*, which as an admixed tree species, strengthens the stand and gives the owner of it a high economic yield.

According to the management principles, the majority of the stands in the study area are restored using the shelterwood method, which does not lead to the establishment of clearings or temporary deforestation. The shelterwood method is therefore fully within the framework of sustainable management [7], and it is the most environmentally friendly option to all of the components of the forest ecosystem as it fully replicates the natural regeneration of generations in the forest stand [7]. The second method is the regeneration by strip, which produces clear cut areas, but due to the area limitations (width and overall size) according to the law [5], it is also considered to be suitable for sustainable management. The overall application of SFM in the area is fully evidenced by the total area of the clear cut areas per year; they reach only units of ha per year, i.e., about 0.005% of the area of the territory, but even these areas are reforested within 2 years.

In terms of age, stands with an age of 170 years are also represented in the area, and the length of the life span for deciduous stands dominated by *Fagus sylvatica* are at the upper limit of the length of the life span according to the law [29], i.e., 120–140 years, while only in stands dominated by *Alnus glutinosa* is it shorter, i.e., 80 years. Even so, the regeneration of stands occurring once every 80–120 years is not a significantly negative factor. In terms of the age structure of the stands (Figure 3), the area of regeneration in the next 2050–2100 years will be smaller, which also represents significantly less forestry activity in the area, and thus less harvesting activity.

The only significant factor remains the actual harvesting of timber, i.e., mainly the movement of forestry equipment, i.e., tractors on tractor logging roads. When one is moving on unsurfaced tractor roads, water and fine soil particles can be concentrated during rainfall and wash into the watercourse, causing higher turbidity and long-term change in the water environment. Similarly, traffic over fords across watercourses also has effects, but here, the action occurs over a period of minutes, and this is only a point effect.

No fertilizers, chemicals, or soil liming are used in the forestry management of the forest stands [66], as these are rich habitats with a predominant soil type of cambisols [6,34]. Thus, these factors cannot even be considered as potential actors.

Construction has significant impacts on the biotopes, e.g., roads, stream bed modification by fortification, stream straightening, or the construction of stone steps in the stream bed. This impact takes place at the time of their construction. The network of the main logging roads has been completed, which was completed in the 1970s and 1980s. At present, the improvement of the forest drainage road network consists mainly in the reconstruction of the existing, mainly unpaved second category drainage roads, or some of the logging roads and dust-free or hard first or second category drainage roads. Proposals for new construction are not currently being considered [47]. Where valley roads were constructed, the watercourse beds were straightened, or longitudinal bank embankment strengthening was undertaken at that time. At present, only spot repairs to the bank embankments are being carried out, which have a minimal impact on the habitat, and thus the population of the species. Species still inhabit the affected watercourse streams, e.g., the Jankovický potok site [23], so it can be argued that the greatest impact on the population occurs during the construction period, and the negative impact then subsides after the construction is completed and the population, if it is affected, returns to its previous abundance. There are no reservoirs in the study area at present and no construction is planned that would cause

the permanent destruction of the population below the reservoir (Holuša unpubl.), thus the threat of reservoir construction is only hypothetical.

Most of the factors identified only have a point effect, i.e., the flow was only affected for a few metres. It is very difficult to assume that the factors have an area effect. None of the factors showed a significant effect on the larval density of *C. heros* in the streams with its occurrence.

According to Murányi et Jović [26], the removal of dead or dying trees, or the removal of old trees, or the cultivation of forests as a renewable energy source and the burning of the brush around the habitat may have a negative effect on the population of *C. heros*. These factors are highly debatable, as the *C. heros* species are not affected in any way by the age of the surrounding stands, let alone by cutting down individual trees [27]. The stands used for fuel production would represent forest stands with very low clearing, unless their extent is extensive in the vicinity of the stream, i.e., from tens to hundreds of hectares, where minimal negative effects on the habitat condition and the population of the species can be expected.

Among the factors, we can also include the occurrence of natural floods, which significantly alter the channel, and thus, the population of the species (two consecutive floods within 5 days were observed in June 2012). The effect of flooding is entirely natural, but it may be exacerbated by increased runoff from the road network. Increased runoff in a watercourse only results in a more pronounced shift of the larvae downstream, and the population may be impacted if the larvae are flushed in large numbers to the village intracity where polluted sewage is discharged into the watercourse, and this may reduce the population.

The categorization of *C. heros* habitats as a category of forests with a special purpose [5] can be considered as a proposal to improve current forest management without designating a separate protected area for *C. heros* conservation [67,68], and thereby, limiting the rights of the owner. Forests where the public interest in improving and protecting the environment or other legitimate interests such as fulfilling the non-productive functions of the forest are superior to the productive functions can also be included in the category of special purpose forests [5]. These are forests that are (a) necessary for the conservation of biodiversity or (b) where other important public interests require a different management method. A separate SMUS should be created with regard to the protection of *C. heros*, which would include forest stands in which the passage of machinery through the stream bed would be prohibited, the path of the harvested timber from the alluvium to the approaching road would be as short as possible, which would be at a certain distance from the watercourse, and above all, there would be an appropriate timing of forestry interventions, especially for harvesting in the winter months.

After the announcement is made, detailed management guidelines [67] would be developed for the spatially distributed units in the forests, including a detailed forest intervention plan that is similar to the one developed for another purpose by Holušová et Holuša [69].

## 5. Conclusions

*C. heros* is included in the national Red Lists in most countries where the species occurs. In the Czech Republic, where the northernmost population occurs, it is listed as endangered (EN).

The species composition and the structure of forest stands in the habitats of *C. heros* correspond to the natural conditions, and the forest regeneration is dominated by regeneration under shelterwood, i.e., the current forest management can be considered to be close to nature. If the current management style is continued, the best conditions for the conservation of the *C. heros* population are created. The only significant phenomena are the tractor logging roads, along which the harvested timber is transported. This transport is then carried out via fords over watercourses or, in times of rainfall, the tractor logging roads

are used to flush fine soil into the watercourses. However, none of the studied forestry factors show a significant effect on the population status of *C. heros*.

An improvement to forest management would be to classify forest stands (habitats) with *C. heros* as a category of forests with a special purpose. A separate SMUS should be created with regard to the conservation of *C. heros*, which would include forest stands, in which more gentle logging and timber transport methods would be used.

In the future, monitoring not only the status of the population in each catchment, but also the status of the forest stands and the status of forest management, remains a major task. This monitoring should be carried out on a cycle of about once every five years.

**Author Contributions:** Conceptualization, O.H.; methodology, O.H. and K.H.; software, O.H. and A.B.; validation, O.H. and A.B.; formal analysis, O.H., K.H. an A.B.; investigation, O.H. and K.H.; resources, O.H. and K.H.; data curation, O.H. and K.H.; writing—original draft preparation, O.H., K.H. and A.B.; writing—review and editing, O.H. and K.H.; visualization, O.H., K.H. and A.B.; supervision, O.H.; project administration, O.H. and K.H.; funding acquisition, O.H. and K.H. All authors have read and agreed to the published version of the manuscript.

**Funding:** This research was funded by Agency for Nature and Landscape Conservation of the Czech Republic of in frame of monitoring NATURA 2000 species.

**Institutional Review Board Statement:** Not applicable.

**Data Availability Statement:** Not applicable.

**Acknowledgments:** We thank to Robert Doležal (Brno) for helping to create map images.

**Conflicts of Interest:** The authors declare no conflict of interest. The funders had no role in the design of the study; in the collection, analyses, or interpretation of data; in the writing of the manuscript; or in the decision to publish the results.

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
