# Peer review of "Is the Current Forest Management to the Northernmost Population of Cordulegaster heros (Anisoptera: Cordulegastridae) in Central Europe (Czech Republic) Threatening?"

_forests, doi:10.3390/f14020228_

Round 1

Reviewer 1 Report

The evaluated paper is a valuable analysis of the occurrence of a species of "special concern" associated with forest areas, against the background of detailed and competently presented parameters of forest management. There are few such publications, because few authors know equally well the ecology of dragonflies and the forestry. It is therefore a study that can be a model for other similar analyses carried out in the future, for other areas and other species, especially of the genus Cordulegaster. It also brings new elements to the knowledge about the threats of Cordulegaster heros. It is also valuable to show that forest management can be planned in such a way that it does not threaten these dragonflies. 

I have inserted minor remarks in the text of the paper. They mainly concern technical matters (spaces, use of abbreviations, use of italics).

While I like the substance of the text as a whole, I have serious critical remarks about the methods and results.

(1) The authors analyze the forests in the research area in detail, but the data on C. heros is very incomplete. I assume these data is presented in a recent publication on the occurrence of C. heros at the edge of its range in the Czech Republic: Diversity 2022, 14(10), 854. This paper is cited several times in the introductory part (e.g. in lines no. 68 and 71) but there is no specific reference in the methodological part. We only have Figure 2 in the subsection "2.1. Study area", where the distribution of the species against the background of the hydrological network of the study area is shown. It is not enough. I think that there should be a separate methodological subchapter devoted to C. heros research. Of course, this can be short and with a reference to the details in the paper in Diversity. But this must be taken into account, because the current important element of the analysis - the distribution and size of the C. heros populations - appears in the reviewed publication, like a deus ex machina.

(2) There is also no data about C. heros in "Results". This chapter encompasses a very detailed analysis of forests, a list of potential threats and then an assessment of their impact on C. heros populations. The lack of data on C. heros means that some results also appear like a deus ex machina. As in the description of the methods, the results may be short and with a reference to the details in the paper in Diversity. But they must appear in some form. Every publication should be self-explanatory.

(3) How was the impact of factors on C. heros analyzed? It seems to me that this requires a statistical analysis: it is necessary to determine the impact and statistical significance of individual factors. There is no information about this in the paper, neither in "Materials and Methods" nor in "Results". This makes the authors' conclusions seem not to be adequately substantiated by the results. I do not rule out that such analyzes have been made. Perhaps they were not shown in the paper so as not to extend it. However, the presentation of the results of such an analysis and its methodological foundations is a sine qua non condition for the credibility of the conclusions.

Author Response

Thank you for your insightful comments. We have accepted all of your suggestions and have significantly supplemented the manuscript with the data you requested.

Reviewer 2 Report

Manuscript entitled “Is the Current Forest Management to the Northernmost Population of Cordulegaster heros (Anisoptera: Cordulegastridae) in Central Europe (Czech Republic) threatening?” presents state of forest stands, management practices and factors that could negatively affect the northernmost population of C. heros. However, I have found some minor mistakes:

Line 156: …to have a stock of 800-1000 m3.

Table 1: I think that Betula pendula is valid name and B. verrucosa synonym. Explain in table caption what + means (probably presence of the species, without relevant share

Lines 180, 181: SMU 41 is mentioned twice, probably some other SMU should be written, maybe SMU 25?

Line 218: what is “oa”?

Line 268 (and many times later in the text, also Figure 10 caption and Table 5): word “cathegory” should be replaced with “category”

Line 306: Threatening factors on the population…

Line 324: The sentence ends unclearly. Maybe …where C. heros is present.

Line 351-353: Explain what category “EN” and “VU” means – the same as you already did for “NT” in the same paragraph.

 For the authors to consider (but it is not necessarily taken into account):

-        an explanation could be added as to why water turbidity threatens the population of C. heros

-       I agree with the authors “stands of natural tree species have a high ecological stability and there is a very low probability that they would undergo extensive decay to cause extensive deforestation of the entire area”. However, is anything threatening these natural tree species? For example, are two important floodplain tree species A. glutinosa and F. excelsior threatened by Phytophthora alni or Hymenoscyphus fraxineus, respectively?

Author Response

Thank you for your review report. All your comments have been accepted and corrected in the text. 

Round 2

Reviewer 1 Report

This is a sedond eview of a text that was already rated as good in the first review. The changes made removed the previously indicated defects. The work can be published in its current form, i.e. without changes.